# Clinical Role of Aspirin in Mood Disorders: A Systematic Review

**DOI:** 10.3390/brainsci9110296

**Published:** 2019-10-29

**Authors:** Qin Xiang Ng, Krishnapriya Ramamoorthy, Wayren Loke, Matthew Wei Liang Lee, Wee Song Yeo, Donovan Yutong Lim, Vivekanandan Sivalingam

**Affiliations:** 1MOH Holdings Pte Ltd., 1 Maritime Square, Singapore 099253, Singapore; 2Institute of Mental Health, Buangkok Green Medical Park, 10 Buangkok View, Singapore 539747, Singapore; 3Fitzwilliam College, University of Cambridge, Storey’s Way, Cambridge CB3 0DG, UK; 4National University Hospital, National University Health System, Singapore 119074, Singapore; 5Yong Loo Lin School of Medicine, National University of Singapore, Singapore 117597, Singapore

**Keywords:** aspirin, anti-inflammatory, psychiatry, mood disorder, depression, bipolar, systematic review

## Abstract

Worldwide, depression and bipolar disorder affect a large and growing number of people. However, current pharmacotherapy options remain limited. Despite adequate treatment, many patients continue to have subsyndromal symptoms, which predict relapse in bipolar illness and often result in functional impairments. Aspirin, a common nonsteroidal anti-inflammatory drug (NSAID), has purported beneficial effects on mood symptoms, showing protective effects against depression in early cohort studies. This systematic review thus aimed to investigate the role of aspirin in mood disorders. Using the keywords (aspirin or acetylsalicy* or asa) and (mood or depress* or bipolar or mania or suicid*), a comprehensive search of PubMed, EMBASE, Medline, PsycINFO, Clinical Trials Register of the Cochrane Collaboration Depression, Anxiety and Neurosis Group (CCDANTR), Clinicaltrials.gov and Google Scholar databases found 13,952 papers published in English between 1 January 1988 and 1 May 2019. A total of six clinical studies were reviewed. There were two randomized, placebo-controlled, double-blind trials and populations drawn from two main cohort studies (i.e., the Geelong Osteoporosis Study and the Osteoarthritis Initiative study). Using a random-effects model, the pooled hazard ratio of the three cohort studies was 0.624 (95% confidence interval: 0.0503 to 1.198, *p* = 0.033), supporting a reduced risk of depression with aspirin exposure. Overall, the dropout rates were low, and aspirin appears to be well-tolerated with minimal risk of affective switch. In terms of methodological quality, most studies had a generally low risk of bias. Low-dose aspirin (80 to 100 mg/day) is safe, well-tolerated and potentially efficacious for improving depressive symptoms in both unipolar and bipolar depression. Due to its ability to modulate neuroinflammation and central nervous system processes, aspirin may also have valuable neuroprotective and pro-cognitive effects that deserve further exploration. Further randomized, controlled trials involving the adjunctive use of aspirin should be encouraged to confirm its therapeutic benefits.

## 1. Introduction

Worldwide, mood disorders (depression and bipolar disorder) are a growing public health concern. Depression alone affects over 350 million people globally and accounts for approximately 7.5% of healthy years lost due to disability [1]. A 2010 Global Burden of Disease Study reported that depression and bipolar disorder resulted in more than 85 million disability-adjusted life years [2], a startling human and economic cost.

Depressive and bipolar disorders are both chronic conditions, and long-term maintenance treatment is necessary to prevent recurrence of symptoms, reduce residual symptoms, and maintain functioning [3,4]. However, pharmacotherapy of mood disorders remains suboptimal, often plagued with persistence of symptoms despite an adequate trial of treatment. Several large-scale meta-analyses have raised questions about the efficacy of selective serotonin reuptake inhibitor (SSRI) antidepressants, which are first-line drugs frequently used to treat depression [5,6]. The management of bipolar disorder remains a clinical challenge [7], with variable responses to treatment depending on the polarity of patients [8], and many patients continue to have subsyndromal symptoms. Subsyndromal symptoms in bipolar illness predict relapse and often result in functional impairments [9]. There is thus a need to research new and more-effective acute and maintenance therapies.

Aspirin, also known as acetylsalicylic acid (ASA), is a nonsteroidal anti-inflammatory drug (NSAID) commonly prescribed for pain, fever as well as the prevention of cardiovascular events (including stroke and myocardial infarction) [10]. Aspirin exerts its therapeutic effects via inhibition of cyclooxygenase (COX)-1 and COX-2 systems, reducing the production of prostaglandin E2 (PGE2) and pro-inflammatory cytokines [11]. Increasing evidence points to a pro-inflammatory state in patients with unipolar depression (increased serum interleukin (IL)-6) [12] and bipolar depression (increased levels of IL-1, IL-6 and TNF-alpha) [13]. As such, ASA may ameliorate systemic and brain inflammation and serve as an effective therapeutic adjunct for patients suffering from mood disorders.

The potential therapeutic benefit of ASA for the prevention of depression was suggested by early analyses of the longitudinal Geelong Osteoporosis Study conducted between 1994 and 1997 [14]. Exposure to ASA was associated with a significantly decreased risk of depression (odds ratio 0.18, 95% confidence interval (CI): 0.02 to 1.39). Newer studies showing supporting or null associations have emerged since then. No systematic review on the topic has been performed. A systematic review of the clinical role of ASA in mood disorders is therefore timely to summarize current evidence and guide further research.

## 2. Methods

### 2.1. Search Strategy and Study Identification

A systematic literature search was performed in accordance with Preferred Reporting Items for Systematic Reviews and Meta-Analyses (PRISMA) guidelines. By using the following combinations of broad Major Exploded Subject Headings (MesH) terms or text words (aspirin or acetylsalicy* or asa) and (mood or depress* or bipolar or mania or suicid*), a comprehensive search of PubMed, EMBASE, Medline, PsycINFO, Clinical Trials Register of the Cochrane Collaboration Depression, Anxiety and Neurosis Group (CCDANTR), Clinicaltrials.gov and Google Scholar databases yielded 13,952 papers published in English between 1 January 1988 and 1 May 2019. Attempts were made to search the grey literature using Google search engine. Title/abstract screening were performed independently by three study investigators (Q.X. Ng, W.R. Loke, and M.W.L. Lee) to identify articles of interest. All retrieved publications were manually reviewed and also checked for references of interest.

### 2.2. Study Selection Criteria and Eligibility Criteria

The inclusion criteria for this review were (1) published clinical study, (2) study participants with diagnosed depressive or bipolar disorder, (3) use of ASA, and (4) available outcome data pertaining to mood symptoms and control. Any disagreement on inclusion was resolved by consensus. Conference abstracts and proceedings were not accepted for inclusion into this systematic review.

### 2.3. Data Extraction and Risk of Bias

Data were extracted using a standardized electronic form by one study investigator (Q.X. Ng) and cross-checked by a second investigator (W.R. Loke) for accuracy.

The Cochrane Collaboration’s tool for assessing risk of bias [15] of randomized, controlled trials and the Newcastle–Ottawa Scale [16] for cohort studies was independently applied by three study investigators (Q.X. Ng, K. Ramamoorthy, and M.W.L. Lee) to assess the quality of the studies reviewed. Any disagreement was resolved by discussion and consensus among the three researchers.

### 2.4. Statistical Analyses

Hazard ratios reported by the different cohort studies were pooled using a random-effects model, assuming that the selected studies are random samples from a larger population. Heterogeneity was examined using the I^2^ statistic and Cochran’s Q test. Publication bias was assessed using a funnel plot and Egger test. All analyses were done using MedCalc Statistical Software version 14.8.1 (MedCalc Software bvba, Ostend, Belgium; 2014).

## 3. Results

The abstraction process (and reasons for exclusion) is summarized in Figure 1. 

A total of six clinical studies were systematically reviewed. The salient details of the studies are summarized in Table 1. There were two randomized, placebo-controlled, double-blind trials and two key cohort studies (the Geelong Osteoporosis Study [14] and the Osteoarthritis Initiative (OAI) study [17]). A meta-analysis was limited due to the small number of studies and dissimilar study designs and outcome measures. For the same reasons, a sensitivity analysis was not performed.

Using a random-effects model, the pooled HR of the three cohort studies [14,17,21] was 0.624 (95% CI: 0.0503 to 1.198, *p* = 0.033), supporting a reduced risk of depression with ASA exposure. The high heterogeneity (I^2^ = 79.25%), as seen in Figure 2, could be due to the different population characteristics studied (e.g., patients being at different phases of illness, of differing demographics, and on different treatments).

In terms of methodological quality, it was encouraging that most studies had a generally low risk of bias (Table 2 and Table 3).

## 4. Discussion

Overall, there is some evidence to support the antidepressant effects of ASA for both unipolar and bipolar depression. The effective management of bipolar disorder remains a clinical challenge. The clinical role of serotonergic antidepressant medication in bipolar depression is still debated and often cautioned due to concerns of manic/hypomanic switch [22]. Importantly, ASA alleviates depressive symptoms and does not appear to induce affective switch or de-stabilization. In a randomized, controlled study of patients with bipolar disorder (*N* = 32) maintained on lithium and ASA (240 mg/day), baseline and endpoint serum lithium concentrations and mood symptoms remained stable throughout the duration of the study [18]. There was no significant difference in mania or depressive symptoms between the group who received aspirin (240 mg/day) or placebo, along with lithium maintenance therapy. In another randomized trial of bipolar patients (*N* = 99) who are at least moderately depressed, there was a main effect of ASA on treatment response, and only one patient who received both ASA (81 mg twice daily) and minocycline (100 mg twice daily) developed hypomania. Overall, the drop-out rate was low and ASA appears well-tolerated with minimal risk of affective switch. 

The pooled HR of the three cohort studies [14,17,21] was 0.624 (95% CI: 0.0503 to 1.198, *p* = 0.033), supporting an overall reduced risk of depression with ASA exposure. The potential therapeutic mechanisms of ASA on mood likely stem from its effects on COX-1 and COX-2 systems, inhibiting the arachidonic acid pathway (which is a central regulator of inflammatory response) and in the process, reducing the production of PGE2 and pro-inflammatory cytokines [11]. Increasing evidence have supported a cytokine hypothesis of depression and the existence of a pro- inflammatory state in patients with unipolar depression (increased serum interleukin IL-6) [12] and bipolar depression (increased levels of IL-1, IL-6 and TNF-α) [13,23]. Animal studies have found that COX-2 inhibition attenuates neuroinflammation (hippocampal inflammatory markers cytokines IL-1β, TNF-α, and brain PGE2 levels) and circulating corticosterone, and may also alleviate symptoms of anxiety and cognitive decline [24]. Moreover, lithium, which is the current gold standard pharmacotherapy for bipolar patients [25], also shares these characteristics. Previous studies have found that it is able to decrease the production of TNF-α, IL-1β, and PGE2 in glial cells [26,27].

In addition to the effects of COX inhibition on the inflammatory cascade, ASA could also modulate central nervous system (CNS) processes. Adult human microglia predominantly express COX-1 [28]. Several lines of evidence suggest the involvement of microglial activation in the pathogenesis of bipolar disorder [23,29]. In this vein, as ASA covalently modifies COX-1, this could at least in part account for its therapeutic effects for patients with bipolar disorder. A recent study conducted using the FAD5X mice model for Alzheimer’s disease also found that ASA binds to peroxisome proliferator-activated receptor alpha (PPARα) and upregulates the expression of brain-derived neurotrophic factor (BDNF) in hippocampal neurons. These actions are beneficial for both cognition and mood as BDNF enhances serotonergic (5-HT) neurons and has a postulated central role in synaptic plasticity and neuroplasticity [30]. It is worth mentioning that cognitive dysfunction is common in patients with mood disorders, and it remains difficult-to-treat and is associated with poorer clinical outcomes and impaired functioning [31]. Some cohort studies have reported that ASA protects against cognitive decline [10,32]. Similarly, a new antidepressant, vortioxetine improves cognitive function by increasing BDNF in the hippocampus [33].

Minocycline, which was administered in one of the trials, along with aspirin [19], is an antibiotic that has anti-inflammatory properties and also acts on microglial cells [34]. It has demonstrated antidepressant effects in clinical studies [35], albeit contradictory results have been reported in animal models [36].

Finally, drug safety is also an important consideration, especially in choosing long-term therapies. In the above studies reviewed, there were no reports of any severe adverse events related to ASA use. Findings from the landmark Japanese Primary prevention of atherosclerosis with Aspirin for Diabetes (JPAD) trial also support the tolerability and safety of low-dose ASA (80 to 100 mg/day) [37]. As hemorrhagic stroke is more common in the Japanese than Western populations [38], it was encouraging that the risk of hemorrhagic stroke was similar between the treatment and control group in the JPAD trial [37]. Moreover, a 10-year follow-up of the patients enrolled in the original JPAD trial found no increased risk of cardiovascular events with low-dose ASA use but slight increased risk (*p* = 0.03) for gastrointestinal bleeding [39].

## 5. Conclusions

In conclusion, there is considerable evidence to support the clinical role of aspirin in the management of mood disorders. Low-dose aspirin (80 to 100 mg/day) appears safe, well-tolerated, and efficacious for improving depressive symptoms and preventing bipolar relapse. Due to its ability to modulate neuroinflammation and CNS processes, aspirin may also have valuable neuroprotective and pro-cognitive effects that deserve further exploration. More randomized, controlled trials involving the adjunctive use of aspirin are warranted to confirm its therapeutic benefits.

## Figures and Tables

**Figure 1 brainsci-09-00296-f001:**
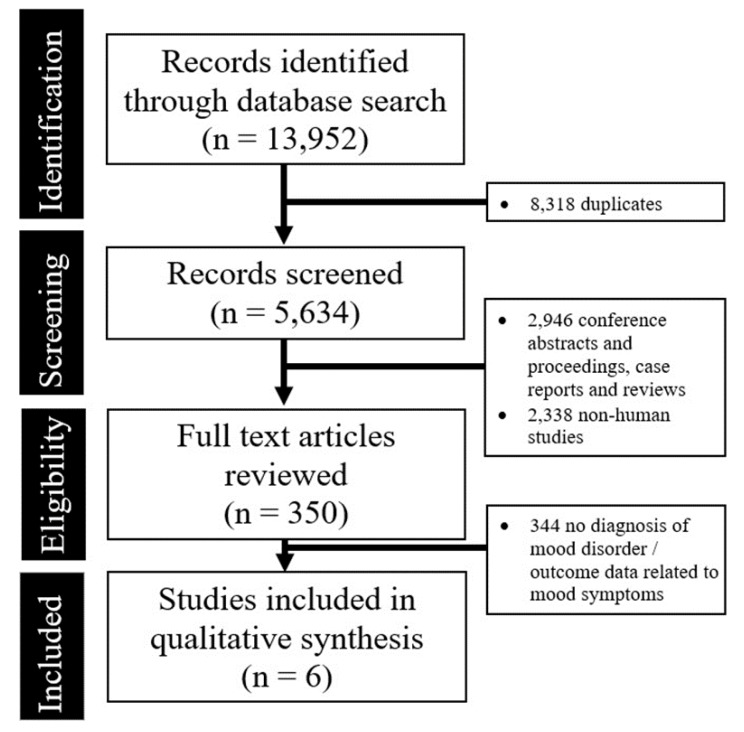
Preferred Reporting Items for Systematic Reviews and Meta-Analyses (PRISMA) flow diagram summarizing the studies identified during the literature search and the abstraction process.

**Figure 2 brainsci-09-00296-f002:**
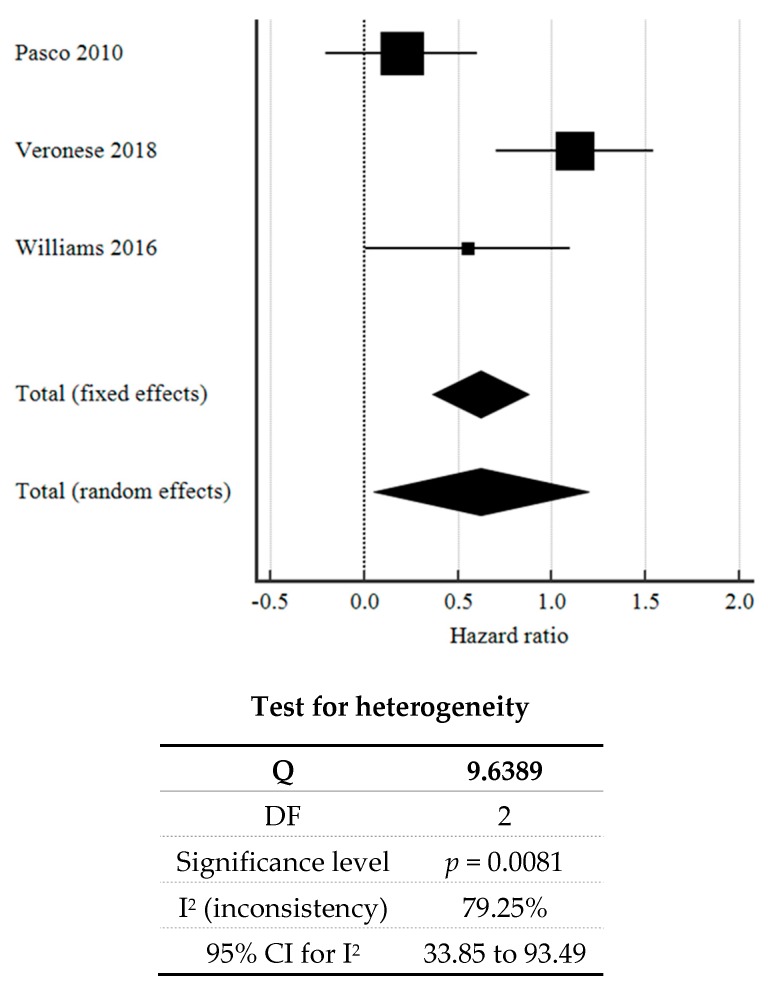
Forest plot showing pooled hazard ratios for the effect of ASA exposure on incident depression.

**Table 1 brainsci-09-00296-t001:** Clinical studies investigating the effects of acetylsalicylic acid (ASA) on mood disorders (arranged alphabetically by first author’s last name).

Author, Year	Study Design	Sample Size (*N*)	Study Population and Duration	Country of Origin	Conclusions
Saroukhani, 2013 [18]	Randomized, placebo-controlled, double-blind	32	Males with stable bipolar affective disorder (DSM-IV-TR) on maintenance lithium therapy; 6 weeks	Iran	Patients who received aspirin (240 mg/day) had significant improvements in total sexual function and erective function domain scores than placebo group. Baseline and endpoint serum lithium concentrations and mood symptoms remained stable throughout the duration of the study.
Savitz, 2018 [19]	Multi-site, randomized, placebo-controlled, double-blind	99	At least moderately depressed psychiatric outpatients with Bipolar I, II or NOS (DSM-IV-TR criteria); 6 weeks	United States	Active minocycline (100 mg twice daily) and aspirin (81 mg twice daily) significantly improved depressive symptoms. There was a main effect of aspirin on treatment response.
Stolk, 2010 [20]	Retrospective linkage record	5145	Patients ≥18 years old, who had been dispensed at least five prescriptions for lithium; 10-year period of observation	Netherlands	Presumably, low-dose ASA (30 or 80 mg/day) significantly reduced the relative risk of clinical deteriorations in patients on lithium (adjusted incidence density of medication events (dose increase or drug change) was 0.84, 95% CI: 0.75 to 0.94).
Pasco, 2010 [14]	*Study 1:* Nested case–control	386	Community-dwelling females; followed for 10 years; population derived from the Geelong Osteoporosis Study	Australia	ASA use associated with protective effect against major depression (age-adjusted OR 0.18, 95% CI: 0.02 to 1.39, *p* = 0.1).
	*Study 2:* Retrospective cohort	345			Reduced risk of major depression in individuals with history of ASA and statin exposure (HR 0.20, 95% CI: 0.04 to 0.85, *p* = 0.03).
Veronese, 2018 [17]	Longitudinal cohort	4070	Community-dwelling adults; followed for 8 years; population derived from ongoing multicenter, longitudinal Osteoarthritis Initiative (OAI) study	United States	Adjusting for confounders, ASA use did not protect against incident depressive symptoms over the study period of 8 years (HR 1.12; 95% CI: 0.78 to 1.62, *p* = 0.54).
Williams, 2016 [21]	*Study 1:* Nested case–control	937	Community-dwelling males, 24–98 years old; followed for 5 years; population derived from ongoing Geelong Osteoporosis Study	Australia	After adjustment for age and antidepressant use, exposure to ASA was associated with a reduced likelihood of major depression (OR 0.4, 95% CI: 0.2 to 0.9, *p* = 0.03).
	*Study 2:* Retrospective cohort	836			Reduced risk of major depression in individuals with history of ASA and statin use (HR 0.55, 95% CI: 0.23 to 1.32, *p* = 0.18).

Abbreviations: ASA, aspirin; DSM-IV-TR, Diagnostic and Statistical Manual of Mental Disorders, Fourth Edition Text Revision; HR, hazard ratio; OR, odds ratio; CI, confidence interval.

**Table 2 brainsci-09-00296-t002:** The Newcastle–Ottawa Scale for assessing the quality of cohort studies reviewed.

Study	Representativeness of the Exposed Cohort ^a^	Selection of the Non-Exposed Cohort ^a^	Ascertainment of Exposure ^a^	Demonstration that Outcome of Interest was Not Present at Start of Study ^a^	Comparability of Cohorts ^b^	Assessment of Outcome ^a^	Follow-up Duration ^a^	Follow-up Adequacy ^a^
Pasco, 2010 (Study 2) [14]	*	*	*	*	**	*	*	*
Veronese, 2018 [17]	*	*	*	*	**	*	*	*
Williams, 2016 (Study 2) [21]	*	*	*	*	**	*	*	*

^a^ A study can be awarded a maximum of one star. ^b^ A maximum of two stars can be given for comparability.

**Table 3 brainsci-09-00296-t003:** Results of Cochrane Collaboration’s tool for assessing risk of bias.

Study (Author, Year)	Sequence Generation	Allocation Concealment	Blinding	Incomplete Outcome Data	Selective Outcome Reporting	Other Bias
Saroukhani, 2013 [18]	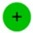	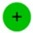	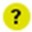	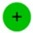	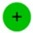	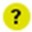
Savitz, 2018 [19]	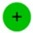	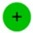	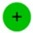	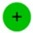	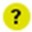	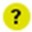

Key: 
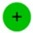
 low risk of bias; 
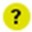
 unclear risk of bias.

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
