# Peer review of "Clinical Role of Aspirin in Mood Disorders: A Systematic Review"

_brainsci, 2019, doi:10.3390/brainsci9110296_

Round 1

Reviewer 1 Report

This systematic review investigated the role of aspirin in mood disorders. A total of six clinical studies were reviewed. There were two randomized, placebo-controlled, double blind trials and populations drawn from two main cohort studies. Using a random-effects model, pooled hazard ratio of the three cohort studies supported a reduced risk of depression with aspirin exposure.

Although small numbers of studies reviewed limited the meta-analysis, this study made a brief review for the effect of aspirin on depression and provided a basis for further study.

Some revisions are needed to improve the manuscript.

“Figure 2. PRISMA flow diagram summarizing the studies identified during the literature search and abstraction process.” Figure 1 or Figure 2? “Test for heterogeneity”: included in Figure 2? “In terms of methodological quality, it was encouraging that most studies had a generally low risk of bias (Table 2 and 3).” Duplicated?

Author Response

"Figure 2. PRISMA flow diagram summarizing the studies identified during the literature search and abstraction process." REPLY: This should be Figure 1. This has now been corrected. “Test for heterogeneity: included in Figure 2?" REPLY: Yes, the test for heterogeneity is considered part of Figure 2. “In terms of methodological quality, it was encouraging that most studies had a generally low risk of bias (Table 2 and 3). Duplicated?" REPLY: The duplicated statement has now been deleted. Apologies for the oversight.

Reviewer 2 Report

This is a comprehensive review that contains interesting information regarding the antidepressant effect of aspirin. It is an useful contribution to the field. The review is in general well written and comprises data on this important topic. Before publication I just have some minor comments: the authors mention in the Discussion also minocycline - here they should specify results suggesting antidepressant efficacy (Rosenblat JD et al., J Affect Disord, 2018), but also preclinical data suggesting rather no implication in depression (Vogt MA et al., Behav Brain Res, 2016).  Also, it should be shortly mentioned what minocycline is (an antibiotic that acts also on microglia).

Author Response

Thank you for the suggestions and references. We have now added the paragraph, "Minocycline, which was administered in one of the trials, along with aspirin [18], is an antibiotic that has anti-inflammatory properties and also acts on microglial cells [35]. It has demonstrated antidepressant effects in clinical studies [36], albeit contradictory results have been reported in animal models [37]." 

Reviewer 3 Report

Well-written review.  Statistics valid.  There is not much in the literature, but is a good review.  No significant changes recommended

Author Response

Thank you for the positive feedback.

Round 2

Reviewer 1 Report

The authors have revised the manuscript adequately. I would like to suggest the editors accept it for publication.